# Role of anterior insula cortex in context-induced relapse of nicotine-seeking

**Hussein Ghareh[1†], Isis Alonso-Lozares[2,3†], Dustin Schetters[2,3], Rae J Herman[2,3], Tim S Heistek[4], Yvar Van Mourik[2,3], Philip Jean-Richard-dit-Bressel[5], Gerald Zernig[1], Huibert D Mansvelder[4], Taco J De Vries[2,3], Nathan J Marchant[2,3]\***

[1]Department of Pharmacology, Medical University of Innsbruck, Innsbruck, Austria; [2]Amsterdam UMC location Vrije Universiteit Amsterdam, Department of Anatomy & Neurosciences, Amsterdam, Netherlands; [3]Amsterdam Neuroscience, Compulsivity Impulsivity and Attention, Amsterdam, Netherlands; [4]Department of Integrative Neurophysiology, Center for Neurogenomics and Cognitive Research, Amsterdam Neuroscience, Vrije Universiteit, Amsterdam, Netherlands; [5]School of Psychology, University of New South Wales, Sydney, Australia

**Abstract** Tobacco use is the leading cause of preventable death worldwide, and relapse during abstinence remains the critical barrier to successful treatment of tobacco addiction. During abstinence, environmental contexts associated with nicotine use can induce craving and contribute to relapse. The insular cortex (IC) is thought to be a critical substrate of nicotine addiction and relapse. However, its specific role in context-induced relapse of nicotine-seeking is not fully known. In this study, we report a novel rodent model of context-induced relapse to nicotine-seeking after punishment-imposed abstinence, which models self-imposed abstinence through increasing negative consequences of excessive drug use. Using the neuronal activity marker Fos we find that the anterior (aIC), but not the middle or posterior IC, shows increased activity during context-induced relapse. Combining Fos with retrograde labeling of aIC inputs, we show projections to aIC from contralateral aIC and basolateral amygdala exhibit increased activity during context-induced relapse. Next, we used fiber photometry in aIC and observed phasic increases in aIC activity around nicotine-seeking responses during self-administration, punishment, and the context-induced relapse tests. Next, we used chemogenetic inhibition in both male and female rats to determine whether activity in aIC is necessary for context-induced relapse. We found that chemogenetic inhibition of aIC decreased context-induced nicotine-seeking after either punishment- or extinction-imposed abstinence. These findings highlight the critical role nicotine-associated contexts play in promoting relapse, and they show that aIC activity is critical for this context-induced relapse following both punishment and extinction-imposed abstinence.

**\*For correspondence:**
n.marchant@amsterdamumc.nl

[†]These authors co-first authors

**Competing interest:** The authors declare that no competing interests exist.

## Editor's evaluation

This manuscript is of interest to readers in the fields of drug addiction and relapse, reinforcement learning and punishment, and those interested in cortical functions, particularly the insular cortex. The authors extend a context and punishment-based relapse model to the widely-used drug nicotine and use a number of carefully controlled complementary approaches ranging from chemogenetic interventions to fiber photometry to support the conclusion that the insular cortex plays a role in nicotine relapse.

## Introduction

Tobacco use is one of the leading causes of preventable death worldwide. In both abstinent and nonabstinent individuals with a history of nicotine use, exposure to cues associated with nicotine use provokes craving (*O'Brien et al., 1992*; *Waters et al., 2004*), which is strongly related to relapse (*Franklin et al., 2007*). Environmental contexts also play a crucial role in nicotine craving. An environmental context associated with nicotine use retains the ability to reinstate cue-induced nicotine craving after extinction in humans (*Bouton and Bolles, 1979*; *Ferguson and Shiffman, 2009*). Preclinical models have been used to study the role of contexts in relapse using the extinction-based context-induced reinstatement (or ABA renewal) model (*Crombag and Shaham, 2002*; *Diergaarde et al., 2008*). One potential limitation of the extinction-based models is that extinction does not capture the motivation for abstinence in humans (*Epstein and Preston, 2003*; *Epstein et al., 2006*). We recently developed a variation of this model in which an alcohol-reinforced response is suppressed by response-contingent punishment (*Marchant et al., 2013*). These studies built on prior models using punishment to model the negative consequences of drug use (*Panlilio et al., 2003*; *Pelloux et al., 2007*; *Vanderschuren and Everitt, 2004*; *Wolffgramm and Heyne, 1995*). We and others have demonstrated context-induced relapse of alcohol, food, and cocaine seeking after punishment-imposed abstinence in an alternative context (*Bouton and Schepers, 2015*; *Farrell et al., 2019*; *Marchant et al., 2016*; *Marchant and Kaganovsky, 2015*; *Marchant et al., 2014*; *Pelloux et al., 2018a*; *Pelloux et al., 2018b*). The extent to which this phenomenon translates to context-induced relapse of nicotine-seeking has not yet been demonstrated.

The insular cortex (IC) has been considered a critical neural substrate of nicotine addiction since it was discovered that some human patients with stroke-induced damage to their insula had a higher probability of smoking cessation (*Naqvi et al., 2007*). Subsequent clinical studies found that nicotine dependence is positively correlated with cue-induced activation in the insula (*Brody et al., 2002*; *Kang et al., 2012*), and there is a negative association between nicotine dependence and insula structural integrity (*Wang et al., 2019*). Insula activity is related to the processing of drug cues (*Yalachkov et al., 2012*), and cue-induced activity in the anterior insula indicates relapse vulnerability (*Gilman et al., 2018*). Both nicotine withdrawal and acute abstinence lead to changes in anterior insula activity, and can also weaken connectivity between the default mode network and salience network at the resting state (*Ding and Lee, 2013*; *Lerman et al., 2014*). In light of these findings, we focus here on the role of the rodent anterior insula cortex (aIC) in context-induced relapse of punished nicotine-seeking.

Here, we demonstrate for the first time, in both male and female rats, context-induced relapse of nicotine-seeking after punishment of nicotine taking in an alternative context. Using the neuronal marker of activity Fos (*Cruz et al., 2013*; *Herdegen and Leah, 1998*; *Morgan and Curran, 1991*), we show that context-induced relapse of punished nicotine-seeking is associated with increased Fos expression in aIC but not middle or posterior IC. We also found that context-induced relapse was associated with increased Fos in projections to aIC from both contralateral aIC and ipsilateral basolateral amygdala (BLA). Next, we used calcium imaging with fiber photometry (*Cui et al., 2013*; *Gunaydin et al., 2014*) to record the activity of aIC neurons throughout nicotine self-administration, punishment, and context-induced relapse. We found that aIC activity was associated with both nicotine infusion and punishment, and also nicotine-seeking during the relapse test. To determine a causal role for activity in aIC and context-induced relapse, we used chemogenetics (*Armbruster et al., 2007*) to inhibit activity in aIC, and found that this decreased context-induced relapse after punishment. Because of potential differences in the neural control of context-induced relapse after punishment or extinction (*Marchant et al., 2019*), we next tested chemogenetic inhibition of aIC after extinction. We also found that this inhibition decreased context-induced reinstatement of nicotine-seeking. These data highlight the critical role that nicotine-associated contexts play in promoting relapse, and they show that activity in aIC is necessary for this, further highlighting a critical role of this structure in relapse to nicotine use.

# Results

## Exp. 1: context-induced relapse to nicotine-seeking after punishment-imposed abstinence is associated with increased Fos expression in aIC, and projections from BLA to aIC

In this experiment, we aimed to identify total activity in IC and BLA, and the inputs to aIC, that are associated with context-induced relapse after punishment. *Figure 1A* shows the experimental timeline. We first injected the rats with the retrograde tracer CTb into aIC, and then trained them to self-administer nicotine via an active nose-poke in one environmental context (context A). An inactive nose-poke response was available but without consequence. After sufficient training, we then punished nicotine taking in an alternative context (context B). We then tested the rats in either context B (Punishment) or context A (Nicotine), or a third group was taken from the home-cage without a final test session (No test). We then processed the brains for immunohistochemical detection of the neuronal marker of activity cFos in combination with CTb.

### Behavioral data

Statistical analysis of the training data (*Figure 1B*) revealed a significant Nose-Poke × Session interaction ($F(14,210) = 16.1$, $p < 0.001$), indicating that responses on the active nose-poke increased throughout training compared to inactive nose-pokes. In punishment (*Figure 1C*), we observed a significant Nose-Poke × Session interaction ($F(6,90) = 9.6$; $p < 0.001$), reflecting the decrease in active nose-pokes during punishment. On the final test, we returned the rats to either context B (Punishment) or context A (Nicotine) (*Figure 1D*). We found a significant main effect of Test Context ($F(1,13) = 14.7$; $p < 0.01$), and a Test Context × Nose-Poke interaction ($F(1,13) = 6.4$; $p < 0.05$). These data show that rats tested in the nicotine context significantly increased nicotine-seeking (active nose-pokes) compared to the rats tested in the punishment context.

### CTb + Fos data

*Figure 1E* shows the spread of CTb at injection site, *Figure 1F* shows example CTb injection in aIC, CTb labeling in pIC and BLA, and CTb + Fos neurons in BLA. *Figure 1G* shows the total Fos, CTb, and percentage CTb + Fos neurons in the insula cortex *ipsilateral* to CTb injection. Using separate one-way analysis of variance (ANOVA) for each region, we found a significant effect of Test Context for Total Fos in aIC ($F(2,17) = 13.1$; $p = 0.001$), mIC ($F(2,17) = 12.6$; $p = 0.001$), and pIC ($F(2,17) = 10.1$; $p = 0.002$). Subsequent Tukey post hoc only revealed a significant difference between rats tested in nicotine versus punishment context for aIC ($p = 0.025$). We found no effect of Test Context for the total CTb in mIC and pIC ($Fs < 1$; $ps > 0.05$). For the percent of CTb neurons that also express Fos, there was a main effect of Test Context in both mIC ($F(2,17) = 5.3$; $p < 0.05$) and pIC ($F(2,17) = 6.5$; $p < 0.05$). However, post hoc analysis revealed no significant difference between the Nicotine and Punishment tested rats.

*Figure 1H* shows the total Fos, CTb, and percentage CTb + Fos neurons in the insula cortex *contralateral* to CTb injection. Using separate one-way ANOVA for each region, we found a comparable pattern of effects for Total Fos. Specifically, there was a main effect of Test Context in aIC ($F(2,17) = 14.1$; $p < 0.001$), mIC ($F(2,17) = 8.3$; $p < 0.01$), and pIC ($F(2,17) = 21.6$; $p = 0.002$), and subsequent post hoc revealed a significant difference between rats tested in nicotine or punishment context in aIC ($p = 0.02$) and pIC ($p = 0.003$), but not mIC ($p = 0.32$). In all three regions we found no effect of Test Context on total CTb ($Fs < 2.5$; $ps > 0.05$). For the percent of CTb neurons that also express Fos, there was a main effect of Test Context in both aIC ($F(2,17) = 7.4$; $p < 0.01$) and mIC ($F(2,17) = 4.2$; $p < 0.05$), but not pIC ($F(2,17) = 1.8$; $p > 0.05$). Post hoc analysis revealed significant difference between the Nicotine and Punishment tested rats in aIC ($p = 0.01$) but not mIC ($p = 0.3$). These data show that context-induced relapse of nicotine-seeking is associated with increased activity in the aIC neurons that project to the contralateral aIC.

In *Figure 1I*, we show the total Fos, total CTb, and the percentage of CTb neurons that are also Fos positive in the BLA *ipsilateral* to the CTb injection. No CTb was observed in the contralateral BLA. Anterior and posterior BLA (aBLA and pBLA, respectively) have distinct connectivity and functions (*Reppucci and Petrovich, 2016*), and were thus analyzed separately. One-way ANOVA revealed a significant effect of Test Context in both anterior BLA ($F(2,17) = 21.1$; $p < 0.001$) and posterior BLA

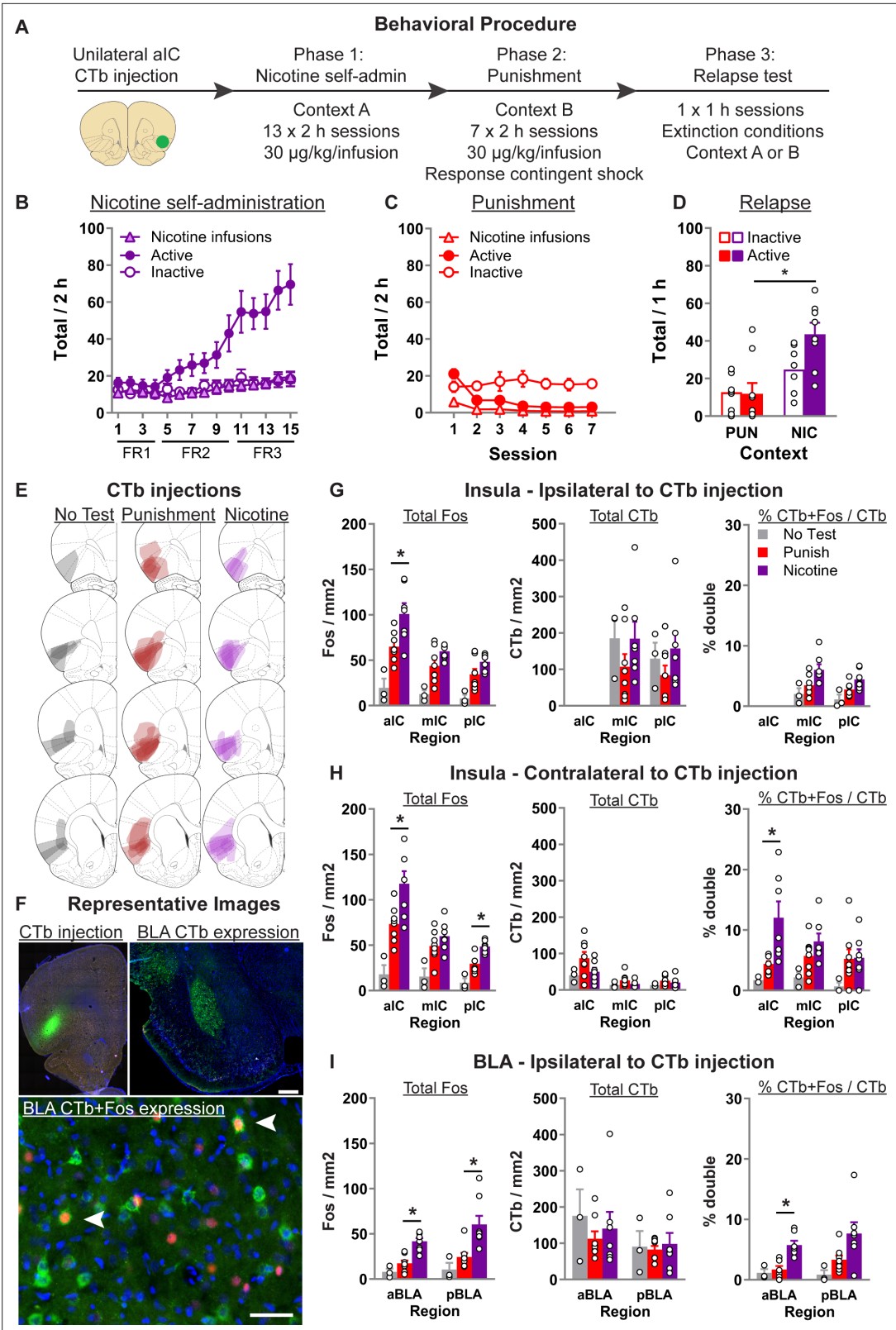

**Figure 1.** Context-induced relapse of punished nicotine-seeking is associated with selective activation of BLA → aIC and contralateral aIC → aIC projections. (**A**) Outline of the experimental procedure (*n* = 19 females). Mean ± standard error of the mean (SEM) active and inactive nose-pokes, and nicotine infusions, during nicotine self-administration in context A (**B**), punishment in context B (**C**), and the context-induced nicotine-relapse test in context B or A (**D**). (**E**) Representative plots of the spread of CTb injections for the three groups. (**F**) Representative images of CTb injection in aIC, and

*Figure 1 continued on next page*

Figure 1 continued

CTb + Fos in BLA. Data are mean ± SEM number of Fos or CTb neurons per mm$^2$, and percentage Ctb + Fos neurons, in the IC hemisphere ipsilateral to CTb injection (**G**), insular cortex (IC) hemisphere contralateral to the CTb injection (**H**), or BLA ipsilateral to the CTb injection (**I**). *p < 0.05; aIC, anterior insula cortex; mIC, middle insula cortex; pIC, posterior insula cortex; BLA, basolateral amygdala; FR, fixed-ratio.

The online version of this article includes the following source data for figure 1:

**Source data 1.** Figure 1 - raw data.

($F$(2,17) = 9.4; p < 0.001). Subsequent Tukey post hoc revealed a significant difference between rats tested in nicotine or punishment context in aBLA (p < 0.001) and pBLA (p = 0.007). We found no effect of Test Context on total CTb in both aBLA ($F$(2,17) < 1; p > 0.05) and pBLA ($F$(2,17) < 1; p > 0.05). Finally, analysis of the percent of CTb-positive neurons that are also Fos positive revealed a significant effect of Test Context in aBLA ($F$(2,17) = 13.5; p < 0.001) and pBLA ($F$(2,17) = 4.6; p < 0.05). Tukey post hoc analysis revealed a significant difference between rats tested in nicotine versus punishment context in aBLA (p = 0.001) but not pBLA (p = 0.1). In summary, these data show that context-induced relapse of nicotine-seeking is associated with increased activity in BLA, and that there is also selectively increased activity in the aBLA → aIC pathway.

## Exp. 2: real-time neuronal activity in aIC encodes nicotine-seeking responses across nicotine self-administration, punishment, and context-induced relapse

In this experiment, we aimed to identify aIC activity during nicotine self-administration, punishment, and relapse. To do this, we used fiber photometry to examine real-time population-level aIC principal neuron calcium (Ca$^{2+}$) transients throughout the entire task. *Figure 2A* shows the experimental timeline. We expressed the Ca$^{2+}$ sensor jGCaMP7f (*Dana et al., 2019*) in aIC using AAV encoding jGCaMP7f under control of the hSyn1 promoter, and measured fluorescence via an optic fiber cannula implanted in aIC (*Figure 2B*). We identified significant Ca$^{2+}$ transients around nose-pokes via bootstrapped confidence intervals and differences between transients via permutation tests controlling for family-wise error rate via a consecutive threshold [0.25 s] and Bonferroni adjustment (*Jean-Richard-Dit-Bressel et al., 2020*).

### Behavioral data

Repeated measures ANOVA on the self-administration data (*Figure 2C*) revealed a significant effect of Nose-Poke ($F$(1,5) = 12.8, p < 0.05). In punishment (*Figure 2D*) repeated measures ANOVA revealed no effect of Nose-Poke ($F$(1,5) = 1.0, p > 0.05). We returned the rats to context A (Nicotine) on the final two test sessions in consecutive days (*Figure 2E*). We found an overall effect of Nose-Poke ($F$(1,5) = 15.8; p < 0.01) reflecting greater responses on the active nose-poke compared to the inactive nose-poke in these test sessions.

### Nicotine self-administration photometry data

In self-administration, we found that aIC has significant excitatory Ca$^{2+}$ transients after nose-pokes. Bootstrapping analysis revealed a significant increase from baseline for nicotine-reinforced active nose-pokes (−0.04 to +8.56 s), nonreinforced active nose-pokes (+0.1 to +5.1 s), and inactive nose-pokes (+0.3 to +4.1 s). The permutation tests comparing the activity of each of these outcomes also revealed significant differences between them (p < 0.01). The most relevant observation is that calcium activity after a reinforced active nose-poke was significantly higher than a nonreinforced active nose-poke (+0.46 to +1.97 s; +2.34 to +7.80 s). This reflected a biphasic excitatory Ca$^{2+}$ response indicating increased activity in aIC related to the nicotine-associated cue and nicotine infusion. Another potentially interesting observation is that there was a significant difference between nicotine-reinforced active and inactive nose-pokes prior to the response (−0.67 s to 0) suggesting a role for aIC in encoding response–outcome contingencies.

*Figure 2—figure supplement 1A* shows these data separately for the FR1, FR2, and FR3 sessions. Bootstrapping analysis revealed a significant increase from baseline for nicotine infusions from the beginning of self-administration and remains throughout. Permutation tests show that the second phase of the biphasic response to nicotine infusions is significantly higher during the FR1 sessions

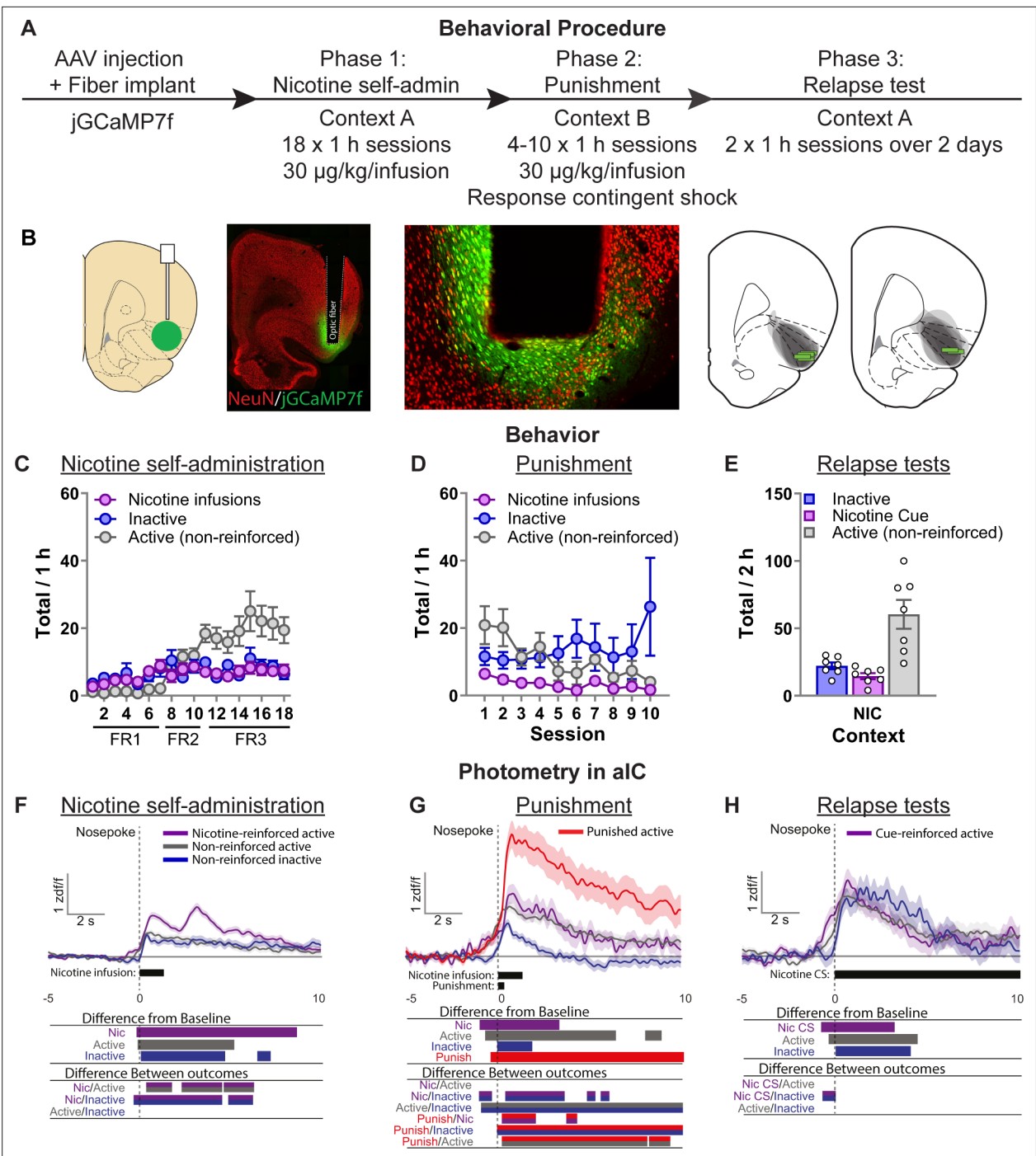

**Figure 2.** Photometry reveals nicotine and punishment-associated activity in anterior insula cortex (aIC). (**A**) Outline of the experimental procedure (*n* = 7 female). (**B**) Representative images of jGCaMP7f expression and fiber implant in aIC. Mean ± standard error of the mean (SEM) nicotine infusions, inactive nose-pokes, and nonreinforced active nose-pokes during nicotine self-administration in context A (**C**), punishment in context B (**D**), and the context-induced nicotine-seeking tests (**E**). (**F**) Ca$^{2+}$ traces around the nose-poke in aIC self-administration in context A (Reinforced active [Nic]: *n* = 751; Nonreinforced active: *n* = 838; Inactive: *n* = 588). (**G**) Ca$^{2+}$ traces around the nose-poke in aIC during punishment in context B (Reinforced active [Nic]: *n* = 89; Nonreinforced active: *n* = 366; Inactive: *n* = 318; Punish: *n* = 137). (**H**) Ca$^{2+}$ traces around the nose-poke in aIC during context-induced relapse test in context A (Active + Nic CS: *n* = 95; Active nonreinforced: *n* = 187; Inactive: *n* = 80). For all photometry traces, bars at bottom of graph indicate significant deviations from baseline (d*F/F* ≠ 0), or significant differences between the specific events (Nicotine infusion, nonreinforced active nose-poke, inactive nose-poke, Punishment, or Nicotine CS), determined via bootstrapped confidence intervals (95% CIs), and permutation tests with alpha 0.008 and 0.01 for comparisons between punishment sessions, and self-administration and tests, respectively. Vertical dashed line indicates nose-poke, horizontal line indicates baseline (d*F/F* = 0).

*Figure 2 continued on next page*

*Figure 2 continued*

The online version of this article includes the following source data and figure supplement(s) for figure 2:

**Source data 1.** Figure 2 - raw data and statistical outputs.

**Figure supplement 1.** Additional photometry analysis across phases.

**Figure supplement 1—source data 1.** Figure 2 - figure supplement 1 statistical outputs.

**Figure supplement 2.** Mean df/f analysis of the photometry data.

**Figure supplement 2—source data 1.** Figure 2 - figure supplement 2 raw data.

(active vs. inactive: +2.5 to +4.0 s), the FR2 sessions (active vs. inactive: +2.41 to +3.4 s), and the FR3 sessions (reinforced active vs. nonreinforced active: +2.72 to +5.6 s). In the FR1 sessions, the first phase of the biphasic response to nicotine (0 to +2.5 s) was not significantly different between active and inactive nose-pokes. This was also the case in the FR2 sessions, where the first phase of the biphasic response to nicotine was not significantly different between nicotine-reinforced and nonreinforced active nose-pokes. Finally, in the FR3 sessions the first phase of the biphasic response to nicotine was significantly higher after nicotine infusion compared to both nonreinforced active and inactive nose-pokes. These data suggest that activity in aIC may be specifically related to outcome processing under uncertainty. Specifically, because as the rats learn the new contingencies in self-administration (from FR1 to FR3) aIC activity decreases to nonreinforced nose-pokes, but remains high for the motivationally salient outcome.

*Figure 2—figure supplement 2A* shows the individual data for the self-administration phase using mean df/f to represent the change in signal from before the nose-poke to after. These data show a significant increase in aIC activity after each response type throughout self-administration.

## Punishment photometry data

In punishment we again found that aIC shows excitatory transients after all nose-pokes. Bootstrapping analysis revealed a significant increase from baseline for unpunished nicotine-reinforced active nose-pokes (+0.08 to +3.28 s) and for nonreinforced active nose-poke (−0.54 to +6.86 s), although comparisons did not indicate a difference between them. We also observed a significant increase after inactive nose-poke (+0.15 s to +1.78). The change in calcium activity to punished active nose-pokes was much more pronounced, lasting the entire epoch of analysis (−0.17 s to +10 s).

Permutation tests comparing the activity of each of these outcomes also revealed some interesting patterns of activity (p < 0.008). First, there was no significant difference between unpunished nicotine-reinforced active nose-pokes and nonreinforced active nose-pokes. This reflects a change from the nicotine self-administration phase, as there was no selective encoding of nicotine infusion compared to nonreinforced active nose-pokes. Second, punished active nose-pokes were significantly higher than both unpunished nicotine-reinforced and nonreinforced active nose-pokes (+0.33 to +10 s), reflecting the profound response of aIC to footshock punishment. As observed during nicotine self-administration training, calcium activity in aIC was significantly higher prior to the nose poke for active responses (nonreinforced, unpunished nicotine, punished nicotine) compared to inactive nose-pokes (−0.73 to 0 s), indicating that aIC activity selectively increases for active nose-pokes prior to the response.

*Figure 2—figure supplement 1B* shows the photometry data from the first punishment session. There are two important differences in this analysis from the overall analysis (*Figure 2G*). First, aIC calcium activity to the nicotine infusion (difference from baseline) is no longer significantly increased. This is a substantial change from nicotine during self-administration, but interestingly the overall analysis (*Figure 2G*) indicates that some degree of activity to nicotine infusions is present in the later punishment sessions. Second is that the response to punishment is far greater in magnitude and duration than both the other responses, as well as during the later punishment sessions. These data further indicate a role for aIC in outcome processing by suppressing the response to a rewarding outcome (i.e., nicotine) when a punishment contingency is introduced. When all punish sessions are combined, activity toward the punished outcome decreases and towards the nicotine increases. This is also consistent with a role for aIC in the integration of information by weighing the value of outcomes.

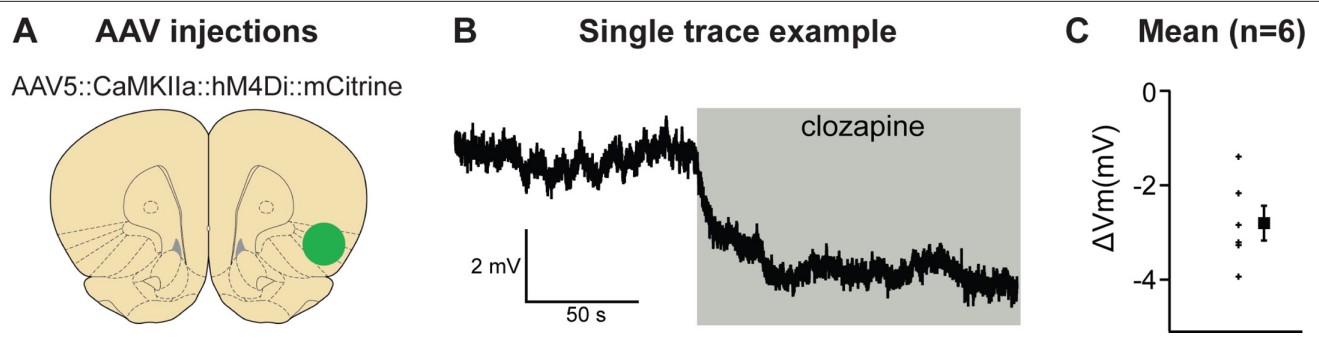

**Figure 3.** Validation of chemogenetic inhibition of anterior insula cortex (aIC) neurons expressing hM4Di by clozapine. (**A**) The virus injection location. (**B**) Example trace of inward current demonstrating clear clozapine-induced hyperpolarization of the hM4Di expressing neurons. (**C**) The average hyperpolarization induced by clozapine for the $n = 6$ neurons recorded (left) and the mean hyperpolarization (right) for these neurons.

The online version of this article includes the following source data for figure 3:

**Source data 1.** Figure 3 - raw data.

*Figure 2—figure supplement 2B* shows the individual data for the punishment phase using mean df/f to represent the change in signal from before the nose-poke to after. These data show a significant increase in aIC activity after each response type throughout punishment.

## Context-induced relapse photometry data

In the final two sessions we tested the rats in the original training context (context A) under extinction conditions over 2 consecutive days. Bootstrapping analysis revealed a significant increase from baseline for active nose-pokes that led to the nicotine-associated cue (−0.54 to +3.41 s), nothing (−0.17 to +4.35 s), and the inactive nose-pokes (+0.21 to +3.97 s). This surprisingly consistent pattern of activity for all types of nose-pokes was reflected in the permutation tests ($p < 0.01$), which revealed no significant difference between the three types of outcomes following the nose-poke. However, like in the previous phases, permutation tests revealed a significant difference between active nose-pokes (leading to nicotine-associated cue) and inactive nose-pokes prior to the response (−0.61 s to 0).

*Figure 2—figure supplement 1C* shows these data separately for the first and second test sessions. This analysis shows that aIC activity prior to an active nose-poke was only significant for the first test session and not the second. We also find a significant difference prior to the nose-poke between nicotine-reinforced active nose-pokes and inactive nose-pokes (−0.54 to 0 s). *Figure 2— figure supplement 2C* shows the individual data for the tests sessions using mean df/f to represent the change in signal from before the nose-poke to after. These data show a significant increase in aIC activity after each response type in these test sessions.

## Exp. 3: validation of chemogenetic inhibition of neuronal activity in AI

We first determined the validity of our chemogenetic inhibition approach and used ex vivo slice physiology in aIC (*Figure 3*). Recordings of hM4Di expressing, mCitrine-positive cells showed that application of 500 nM clozapine (CLZ) hyperpolarized the membrane potential by $2.8 \pm 0.4$ mV ($p < 0.001$).

## Exp. 4: chemogenetic inhibition of aIC decreases context-induced relapse of punished nicotine-seeking

Given the observations of increased Fos in aIC associated with context-induced relapse, and task-related activity in aIC, we aimed to determine whether activity in aIC is necessary for context-induced relapse after punishment-imposed abstinence. *Figure 4A* shows the experimental timeline. In this experiment, rats were injected with AAV encoding either the inhibitory chemogenetic receptor hM4Di or control GFP. We then trained them to self-administer nicotine in context A and then punished nicotine taking in context B. We then tested the rats in both context B (Punishment) or context A (Nicotine) after systemic injection of CLZ (0.3 mg/kg).

*Figure 4B* shows nicotine self-administration in context A. We observed a significant Nose-Poke × Session interaction ($F_{(12,288)} = 13.4$, $p < 0.001$), with responses on the active nose-poke increasing

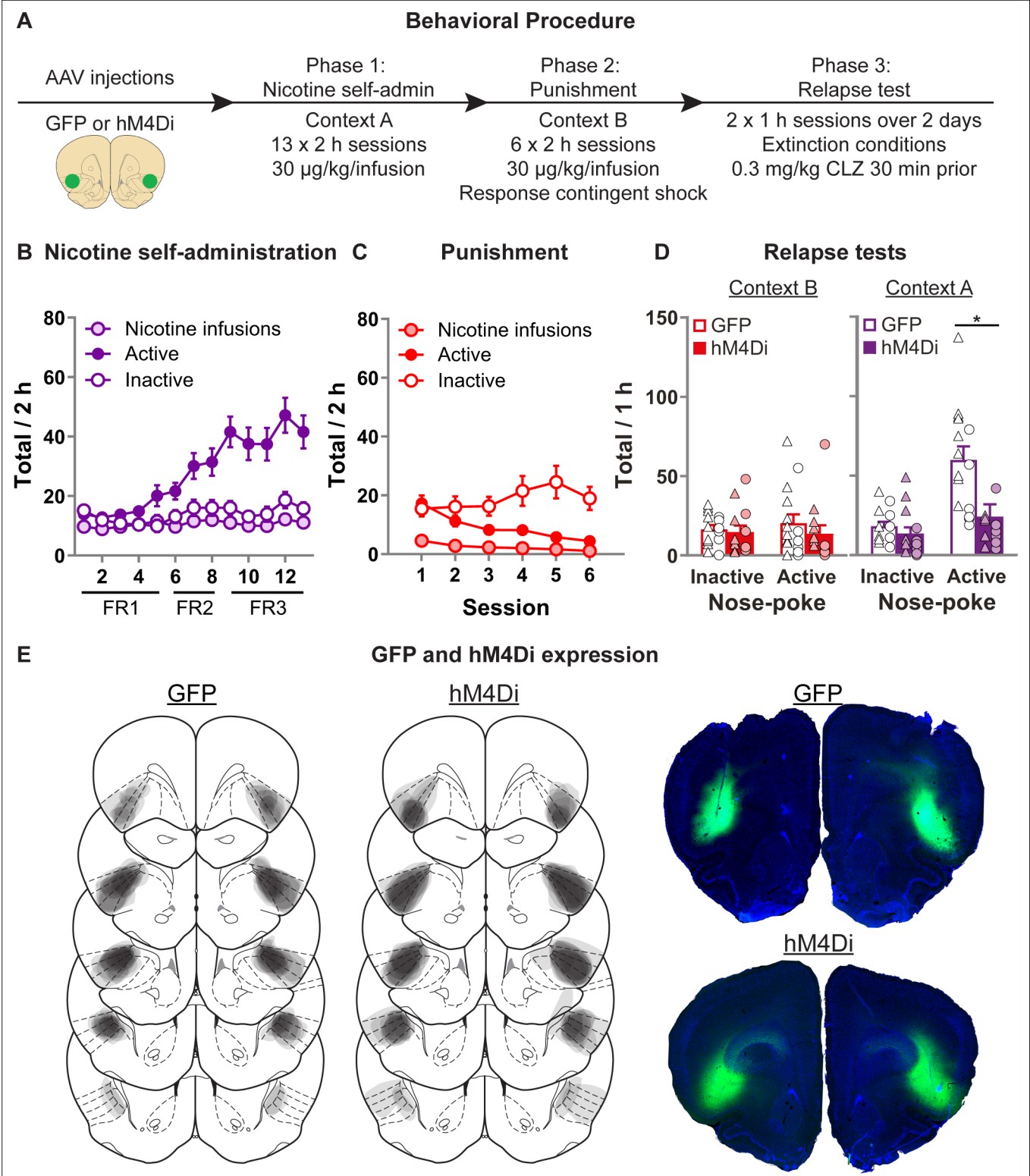

**Figure 4.** Effect of chemogenetic inhibition of anterior insula cortex (aIC) on context-induced relapse of punished nicotine-seeking. (**A**) Outline of the experimental procedure (*n* = 15 females, 13 males). (**B**) Mean ± standard error of the mean (SEM) nicotine infusions, active, inactive nose-pokes during nicotine self-administration in context A. (**C**) Mean ± SEM nicotine infusions, active, inactive nose-pokes during punishment in context B. (**D**) Mean ± SEM nose-pokes during the context-induced relapse tests. Individual data also plotted, triangles = female, circles = male. (**E**) Representative plots of the

*Figure 4 continued on next page*

*Figure 4 continued*

spread of GFP (left) and hM4Di (middle) in aIC of rats in experiment 4. Right top shows an example section of a rat showing GFP expression in aIC, and right bottom shows an example of hM4Di expression. *p < 0.05; CLZ, clozapine; FR, fixed-ratio.

The online version of this article includes the following source data and figure supplement(s) for figure 4:

**Source data 1.** Figure 4 - raw data.

**Figure supplement 1.** Effect of chemogenetic inhibition of anterior insula cortex (aIC) on latency to the first nose-poke during context-induced relapse tests.

**Figure supplement 1—source data 1.** Figure 4 - figure supplement 1 raw data.

throughout training compared to inactive nose-pokes. *Figure 4C* shows nicotine self-administration during punishment in context B. We observed a significant Nose-Poke × Session interaction ($F(5,120)$ = 7.1, p < 0.001), with responses on the active nose-poke decreasing throughout punishment compared to inactive nose-pokes. *Figure 4D* shows nicotine-seeking during the relapse tests. We observed a significant Group × Nose-Poke interaction ($F(1,24)$ = 13.2; p = 0.001), and a Context × Group × Nose-Poke interaction ($F(1,24)$ = 11.6; p = 0.001). We observed no overall effect of Sex ($F(1,24)$ = 4.0; p > 0.05). Moreover, we found no Sex × Group × Nose-Poke interaction ($F(1,24)$ = 2.5; p > 0.05), nor Sex × Context × Group × Nose-Poke interaction ($F(1,24)$ = 3.6; p > 0.05). These results show that chemogenetic inhibition of aIC significantly decreased context-induced relapse of nicotine-seeking after punishment, in both male and female rats.

We conducted two follow-up statistical tests specifically on the inactive nose-pokes to further test for potential nonspecific reductions in behavior caused by the off target effects of CLZ. First we ran a post hoc repeated measures ANOVA on the inactive nose-pokes, and found no Context × Group interaction ($F(1,24)$ < 1; p > 0.05). Next we analyzed the data from the latency to the first nose-poke on the inactive nose-poke during the test sessions (*Figure 4—figure supplement 1*), and found no main effect of Group ($F(1,24)$ = 1.4; p > 0.05), nor Context × Group interaction ($F(1,24)$ < 1; p > 0.05). While indirect, each of these measures indicates that the observed reduction in active nose-pokes does not occur alongside a broader, nonspecific reduction in behavioral output.

## Exp. 5: chemogenetic inhibition of aIC decreases context-induced relapse of extinguished nicotine-seeking

In this experiment, we aimed to determine whether activity in aIC is necessary for context-induced relapse after extinction. *Figure 5A* shows the experimental timeline. We first injected the rats with AAV encoding either the inhibitory chemogenetic receptor hM4Di, or control GFP. We then trained them to self-administer nicotine in context A and then extinguished nicotine-seeking in context B. We then tested the rats in both context B (Extinction) or context A (Nicotine) after systemic injection of CLZ (0.3 mg/kg).

*Figure 5B* shows nicotine self-administration in context A. We observed a significant Nose-Poke × Session interaction ($F(12,324)$ = 12.3, p < 0.001), indicating that responses on the active nose-poke increased throughout training compared to inactive nose-pokes. *Figure 5C* shows nicotine-seeking during extinction in context B. We observed a significant effect of Session ($F(7,189)$ = 14.0; p < 0.001), and Nose-Poke ($F(1,27)$ = 15.6; p < 0.001), but no Session × Nose-Poke interaction ($F(7,189)$ = 1.5; p > 0.05); both the active and inactive nose-poke decreased throughout extinction. *Figure 5D* shows nicotine-seeking during the relapse tests. We observed a significant Group × Nose-Poke interaction ($F(1,27)$ = 8.5; p < 0.01), as well as a Context × Group × Nose-Poke interaction ($F(1,27)$ = 11.3; p < 0.01). We observed no overall effect of Sex ($F(1,27)$ = 1.6; p > 0.05), and no Sex × Group × Nose-Poke interaction ($F(1,27)$ = 1.1; p > 0.05), nor Sex × Context × Group × Nose-Poke interaction ($F(1,24)$ < 1; p > 0.05). These results show that chemogenetic inhibition of aIC significantly decreases context-induced relapse of nicotine-seeking after extinction, in both male and female rats.

We again conducted two follow-up statistical tests specifically on the inactive nose-pokes to further test for potential nonspecific reductions in behavior caused by the off target effects of CLZ. First we ran a post hoc repeated measures ANOVA on the inactive nose-pokes, and found no Context × Group interaction ($F(1,27)$ = 1.5; p > 0.05). Next we analyzed the data from the latency to the first nose-poke on the inactive nose-poke during the test sessions (*Figure 5—figure supplement 1*), and found no main effect of Group ($F(1,27)$ < 1; p > 0.05), nor Context × Group interaction ($F(1,27)$ < 1; p > 0.05).

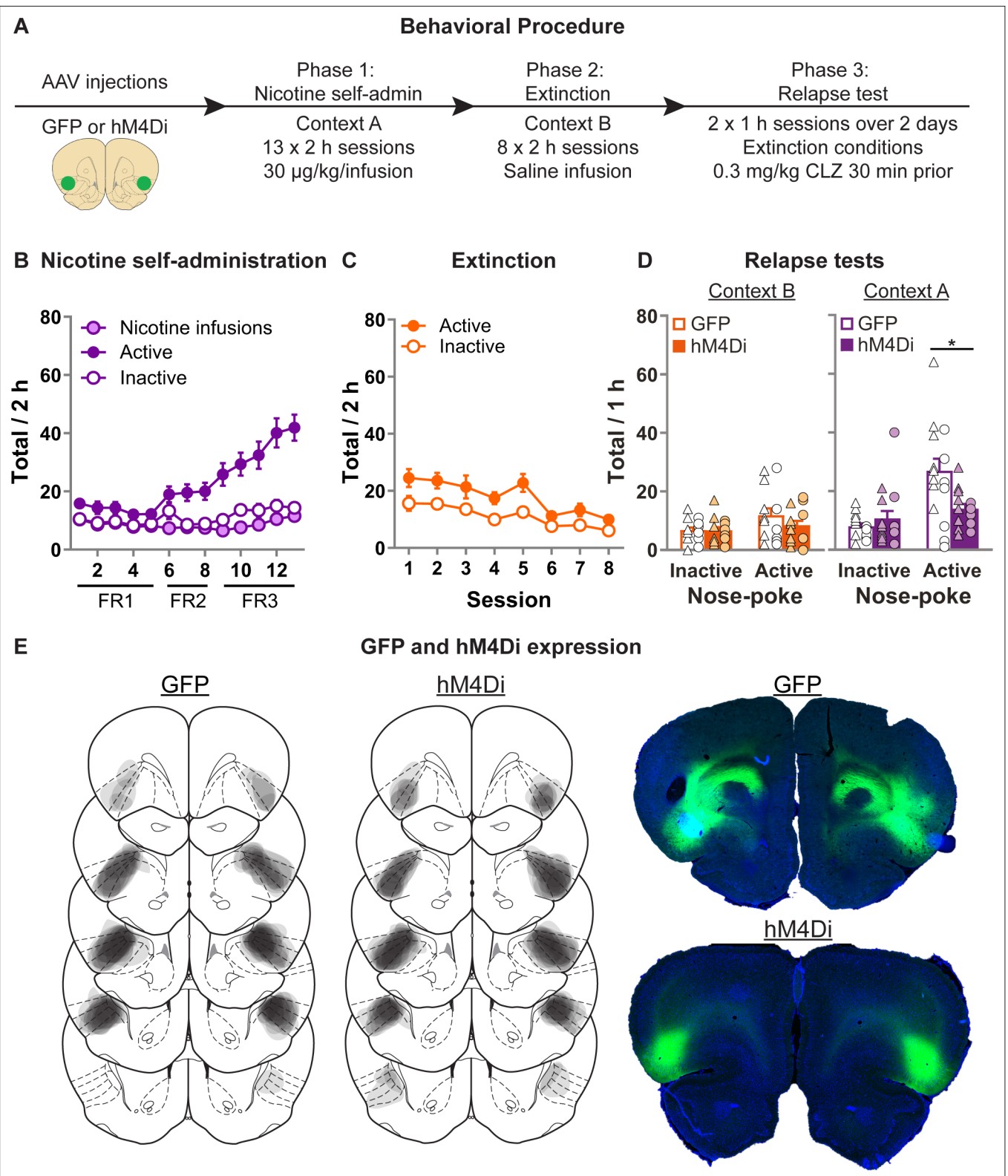

**Figure 5.** Effect of chemogenetic inhibition of anterior insula cortex (aIC) on context-induced relapse of extinguished nicotine-seeking. (**A**) Outline of the experimental procedure (*n* = 18 females, 13 males). (**B**) Mean ± standard error of the mean (SEM) nicotine infusions, active, inactive nose-pokes during nicotine self-administration in context A. (**C**) Mean ± SEM active, inactive nose-pokes during extinction in context B. (**D**) Mean ± SEM nose-pokes during the context-induced relapse tests. Individual data also plotted, triangles = female, circles = male. (**E**) Representative plots of the spread of GFP

*Figure 5 continued on next page*

*Figure 5 continued*

(left) and hM4Di (middle) in aIC of rats in experiment 5. Right top shows an example section of a rat showing GFP expression in aIC, and right bottom shows an example of hM4Di expression. CLZ, clozapine; FR, fixed-ratio.

The online version of this article includes the following source data and figure supplement(s) for figure 5:

**Source data 1.** Figure 5 - raw data.

**Figure supplement 1.** Effect of chemogenetic inhibition of anterior insula cortex (aIC) on latency to the first nose-poke during context-induced relapse tests.

**Figure supplement 1—source data 1.** Figure 5 - figure supplement 1 raw data.

While indirect, each of these measures indicates that the observed reduction in active nose-pokes does not occur alongside a broader, nonspecific reduction in behavioral output.

## Discussion

In this study, we describe a novel rodent model of context-induced relapse to nicotine-seeking after punishment-imposed abstinence. In female rats, we found that this form of relapse is associated with increased Fos expression in aIC, but not mIC or pIC, as well as BLA. Using retrograde tracing from aIC, we also show that inputs from contralateral aIC and ipsilateral anterior BLA are also activated during context-induced relapse of nicotine-seeking. Using fiber photometry in female rats, we found that nicotine infusions during self-administration elicited phasic increases in aIC activity. During punishment, phasic increases in aIC activity were significantly greater for the punishment outcome compared to nicotine infusion and nonreinforced responses. During the final tests, we found increased activity associated with each response type. Interestingly, we also found that aIC activity increased prior to active but not inactive nose-pokes across each phase of the experiment, indicating a potential role of aIC in anticipating outcomes prior to actions, and/or selectively promoting reinforced responses. Next, we used chemogenetics in both male and female rats to show that inhibition of aIC decreased context-induced relapse of nicotine-seeking after both punishment-imposed abstinence and after extinction, in both sexes. This study demonstrates the importance of nicotine-associated contexts in promoting relapse. We show that the aIC activity is critical for this effect, regardless of the mode of abstinence, further highlighting the important role of aIC in relapse of nicotine use.

### Methodological considerations

Several issues must be considered in the interpretation of these findings. In this study, we used intravenous nicotine during self-administration. While human nicotine use is primarily through cigarette smoking, intravenous administration of nicotine is reinforcing in humans (*MacLean et al., 2021*), demonstrating that a comparable route of administration supports reinforcement in humans. In addition, smokeless tobacco is also addictive and is responsible for many adverse health consequences (*Nethan et al., 2018*). Recently, it has been shown that vapor administration of nicotine is effective in rodents (*Lallai et al., 2021*; *Smith et al., 2020*), and vapor exposure causes both physical (*Montanari et al., 2020*) and psychological effects (*Lallai et al., 2021*; *Flores et al., 2022*). The extent to which the route of self-administration changes the neurobiological substrates of context-induced nicotine-seeking is unknown, thus it will be of interest in future studies to determine this. However, because our focus is on the neurobiological substrates by which contexts associated with nicotine or punishment control nicotine-seeking in this study, we argue that the route of administration is unlikely to change the contribution of the aIC to these behaviors.

In the chemogenetics experiments, we used CLZ instead of clozapine-*N*-oxide (CNO) because CNO is converted to CLZ in vivo (*Gomez et al., 2017*), and made comparisons to the GFP group who also received CLZ. Various studies have estimated the conversion ratio of CNO to CLZ in rodents to be between 7.5% and 13% (*MacLaren et al., 2016*; *Manvich et al., 2018*), which leads to an estimate dose used here of 5–10 mg/kg CNO, a well-established CNO dose with minimal side effects (*Mahler and Aston-Jones, 2018*; *Smith et al., 2016*). We found no effect of manipulations on inactive nose-pokes or response latency, suggesting the effect of aIC inactivation is selective nicotine-seeking. However, in the absence of a vehicle control test session it is still not possible to completely rule out nonspecific effects of CLZ on nicotine-seeking. Finally, it is unlikely that hM4Di expression in the

absence of ligand binding will change the function of aIC given the low basal activity of this receptor (*Armbruster et al., 2007*; *Urban and Roth, 2015*).

We used different promoters for viral-induced expression of calcium indicator (jGCamP7f: hSyn) and chemogenetic inhibition (hM4Di: mCaMKIIa). Therefore, the observations reported in the photometry experiment likely include activity from a population of neurons that were not manipulated in the chemogenetics experiments. It will be of interest in future studies to identify potential variation in the responses of aIC neuronal subpopulations to nicotine infusions and punishment. Finally, in this experiment we cannot determine whether the change in aIC activity in response to the various outcomes in this experiment were dependent on a preceding action (i.e., nose-poke), or whether noncontingent outcomes would yield the same result. It will be of interest to measure aIC activity during noncontingent shock delivery, nicotine infusions, and nicotine-associated cues.

## Role of aIC, BLA, and BLA inputs to aIC in context-induced relapse to nicotine-seeking

In preclinical studies, the role of aIC in drug-seeking and relapse is well established (*Arguello et al., 2017*; *Campbell et al., 2019*; *Cosme et al., 2015*; *Pushparaj et al., 2015*; *Venniro et al., 2017*). Bilateral electrical stimulation of the insula, at the level of mIC in this study, has been shown to decrease nicotine self-administration and both cue- and priming-induced reinstatement after extinction (*Pushparaj et al., 2013*). Inactivation of both aIC and mIC can decrease nicotine self-administration, and both drug- and cue-induced reinstatement of extinguished nicotine-seeking (*Pushparaj et al., 2015*; *Forget et al., 2010*). Activity in aIC is also necessary for relapse to alcohol seeking, particularly relapse in the punishment context after a period of extended home-cage abstinence (*Campbell et al., 2019*). Here, we demonstrated a role for aIC in context-induced relapse to nicotine-seeking after both punishment- and extinction-imposed abstinence, further demonstrating the critical role of the aIC in the control of drug seeking.

Our results also show increased activity in BLA during context-induced relapse after punishment-imposed abstinence. The role of BLA in cue- and stress-induced reinstatement of nicotine-seeking has been demonstrated previously (*Khaled et al., 2014*; *Sharp, 2019*; *Xue et al., 2017*; *Yu and Sharp, 2015*), but to our knowledge this is the first time that BLA has been implicated in context-induced relapse of nicotine-seeking. We also show increased activation in the aBLA → aIC pathway during context-induced relapse. Anterior and posterior BLA have different output projections (*Reppucci and Petrovich, 2016*). Inactivation of aBLA decreases reinstatement of extinguished cocaine seeking (*Kantak et al., 2002*), and inhibition of pBLA has been shown to potentiate food and alcohol seeking (*McLaughlin and Floresco, 2007*; *Millan et al., 2015*), as well as impair the expression of punishment leading to increased responses on a punished response (*Jean-Richard-Dit-Bressel and McNally, 2015*). In this study, we show that overall Fos is higher in both aBLA and pBLA during context-induced relapse, but only aBLA projections to aIC had significantly greater activation during relapse, providing further evidence for a functional distinction between anterior and posterior BLA. Projections from BLA to aIC are critical for the maintenance of rewarding contextual stimuli (*Gil-Lievana et al., 2020*). It has been proposed that the more posterior IC regions contribute strongly to the function of the aIC (*Mesulam and Mufson, 1982*). However, we did not observe any increased activity in mIC → aIC or pIC → aIC neurons (either ipsi- or contralateral) in rats tested for context-induced relapse. Given that in this experiment, the nicotine-associated context promotes nicotine-seeking, we propose that the motivational significance of the nicotine-associated context is likely mediated through increased activity in aBLA → aIC neurons and not mIC or pIC neurons.

We also found increased Fos in contralateral aIC projections during context-induced relapse. Corticocortical pathways are primarily thought to result in feedforward inhibition through targeting of the PV interneurons in the contralateral hemisphere (*Anastasiades et al., 2018*; *Carson, 2020*). The contribution of contralateral corticocortical projections to the functions of the frontal cortex is poorly understood, and the importance of this pathway in the regulation of context-induced nicotine-seeking studied is likewise undetermined.

## Role of aIC in the regulation of nicotine taking, punishment, and seeking

Our results using fiber photometry revealed increased aIC neural activity after nose-pokes in all phases of the task. While this suggests that aIC activity may be generally important in encoding response–outcome contingencies, we observed important differences in the patterns of activity in the different phases. In self-administration, aIC activity was highest following nose-pokes that lead to nicotine infusions. During punishment, the response to nonreinforced active nose-pokes changed such that there was no longer a difference between nicotine reinforced nose-pokes and nonreinforced active nose-pokes. This may reflect increased uncertainty of the outcome, as active nose-pokes during this phase could lead to nicotine infusion or nicotine infusion plus shock. Alternatively, it may reflect a change in the motivational determinant of behavior from appetitive to aversive. Indeed, aIC activity in response to the punished nose-pokes was significantly higher than all other outcomes. Such a change in the response of aIC to (nonpunished) nicotine infusion in punishment may reflect a reevaluation of nicotine reward encoding within aIC because the punishment overcomes the motivation for nicotine and suppresses nicotine-seeking. The observed reduction in nicotine-seeking in punishment is an adaptive response, thus we propose that during operant behavior aIC activity may be involved in the adaptation of behavior in response to both rewarding (i.e., nicotine) and aversive outcomes (i.e., punishment). It will be of interest in future studies to determine whether maladaptive responses to punishment, such as the punishment-resistance phenotype (*Marchant et al., 2018*), are associated with differences in the response to either reward or punishment in aIC. Interestingly, previous studies in alcohol trained rats have identified a critical role for aIC in punished alcohol seeking (*Seif et al., 2013*).

The IC is known to support a general function of integrating interoceptive information (*Paulus and Stewart, 2014*), which can play an important role in addictive behaviors. Interoceptive information and external signals from environmental cues converge at the anterior insula (*Craig, 2003*; *Livneh et al., 2020*). In human clinical studies, insula activity is related to both positive and negative emotional reactivity. For example nicotine-associated cue exposure increases aIC activity in nicotine addicted individuals (*Kang et al., 2012*; *Gilman et al., 2018*; *Janes et al., 2010*; *Janes et al., 2017*), and aversive motivational states associated with short-term nicotine withdrawal are also linked to changes in resting state functional connectivity between the insula and associated brain regions (*Fedota et al., 2018*; *Ghahremani et al., 2021*). Furthermore in nonaddicted humans, insula activity is associated with punishment in a risky decision-making task (*Von Siebenthal et al., 2020*). Positive and negative valence signals are integrated in the insula to guide motivated behavior through increased activity in the divergent outputs of insula to various brain regions (*Paulus and Stewart, 2014*; *Livneh et al., 2020*; *Gehrlach et al., 2019*; *Gehrlach et al., 2020*; *Shi and Cassell, 1998*; *Wang et al., 2018*; *Reynolds and Zahm, 2005*; *Kim and Lee, 2012*; *Nicolas et al., 2021*). It will be of interest in future studies to determine whether the activity related to both positive (nicotine infusion) and negative (punished nicotine infusion) outcomes recorded at the population level in our study are selectively encoded through different output pathways of aIC.

We also found that aIC activity increased relative to baseline prior to an active nose-poke, but not inactive nose-pokes. This pattern of activity was consistent throughout the experiment. Other studies have shown that activity in aIC is necessary for the performance of goal-directed behavior (*Parkes and Balleine, 2013*; *Parkes et al., 2015*), but it is not necessary for initial acquisition (*Parkes et al., 2016*). We propose that the differential activity in aIC prior to active versus inactive nose-poke responses may indicate that aIC contributes to the encoding of expectations in operant behavior. This may be consistent with a broader role of IC in the prediction of bodily states (*Livneh et al., 2020*; *Livneh and Andermann, 2021*).

The observation that aIC activity in the relapse tests did not specifically relate to the response type or outcome. While this may be potentially inconsistent with the finding that chemogenetic inhibition decreases context-induced relapse, we think that the use of contextual cues to induce reinstatement may explain this. Photometry analyses are better suited to detecting phasic, rather than background changes in activity, which may be why we did not identify a specific signal related to relapse in the aIC photometry data. Fos studies are better suited for this, and we did observe increased Fos in aIC during context-induced relapse. Our aim in this study was to keep the behavioral task consistent across experiments; it will be of interest in future studies to test whether cue-induced reinstatement

(*Forget et al., 2010*), or trial-based nicotine-seeking using discriminative stimuli (*Cervo et al., 2013*), may more clearly reveal real-time aIC activity patterns associated with reinstatement. However, while relatively small, the significant difference in aIC calcium activity prior to active nose-pokes compared to inactive nose-pokes may also be of relevance here. Given that chemogenetic inhibition of aIC decreased active nose-pokes during the relapse tests, this small change in aIC calcium activity prior to the response may be a critical component of the initiation of nicotine-seeking responses, including the increased seeking within context-induced relapse.

## Similarities and differences in the neural control of relapse after punishment versus extinction

Distinct learning mechanisms are responsible for behavioral control after extinction or punishment. Both are mediated by new context-dependent associations (*Bouton and Schepers, 2015*; *Bouton, 2002*). However, punishment learning involves the acquisition of an association between the response and a novel outcome (shock), while extinction involves the acquisition of an association between the response and no outcome. Here, we show that bilateral chemogenetic inhibition of aIC decreases context-induced relapse of both punished and extinguished nicotine-seeking. Previous studies investigating relapse of either alcohol or cocaine seeking have identified differences in the mechanisms of relapse after punishment or extinction. For example, while dopamine receptor activation in nucleus accumbens core is critical for context-induced relapse of punished alcohol seeking (*Marchant and Kaganovsky, 2015*; *Marchant et al., 2014*), no increase in Fos was observed in nucleus accumbens core after context-induced relapse of extinguished alcohol seeking (*Hamlin et al., 2007*; *Marchant et al., 2009*). Inactivation of BLA potentiates cocaine seeking after punishment, but decreases cocaine seeking after extinction (*Pelloux et al., 2018b*). Meanwhile, inactivation of central amygdala had no effect on cocaine seeking after punishment, but decreased cocaine seeking after extinction (*Pelloux et al., 2018b*). We propose that this finding demonstrates the importance of the aIC in context-induced relapse, regardless of the method used to impose abstinence. It has previously been demonstrated that inhibiting aIC decreases relapse of methamphetamine seeking after choice-based voluntary abstinence (*Venniro et al., 2017*). As such the role of aIC in relapse is likely broader than just for context-induced relapse of nicotine-seeking. It will be of interest in future studies to determine whether this distinction holds true for other drugs of abuse, or indeed for other types of voluntary abstinence such as choice for social reward (*Venniro et al., 2018*).

## Concluding remarks

We sought to investigate the neural substrates of context-induced relapse to nicotine-seeking after punishment-imposed abstinence. Our results show that activity in aIC is necessary for context-induced relapse of punished nicotine-seeking, and this is also the case for extinguished nicotine-seeking. We also show that the BLA projections to aIC are activated during context-induced relapse, and future studies are needed to determine the extent to which this activity is necessary for this relapse. Our findings further highlight the critical importance of the anterior IC as a target for nicotine addiction treatments.

# Materials and methods
## Subjects

We obtained 91 Wistar rats (29 males and 62 females), aged 10–12 weeks upon arrival, from Charles River Laboratories B.V. (Leiden, The Netherlands). In compliance with Dutch law and Institutional regulations, all animal procedures were approved by the Centrale Commissie Dierproeven (CCD; AVD114002016759) and conducted in accordance with the Experiments on Animal Act. Experiments were approved by the local animal welfare body Animal Experiments Committee of the Vrije Universiteit, Amsterdam, The Netherlands. Behavioral tests were conducted during the dark phase of the rat's diurnal cycle (12 hr/12 hr). Food and water were available ad libitum, and rats were single housed the rats after surgery for the remainder of the experiment.

We did not make a specific power analysis to determine sample size prior to any experiments. The group size was chosen based on our past research (*Diergaarde et al., 2008*), suggesting that it will be sufficient to observe significant effects of the role of context on nicotine-seeking. Each experiment

is comprised of data from at least one replication cohort, and cohorts were balanced by viral group, sex, prior to the start of the experiment. We allocated the rats randomly to one of the groups within each experiment, but we were not blinded to the specific group because we were required to administer virus. We did not exclude any rats for reasons of behavioral variation (i.e., no outliers have been removed), but rats that did not have correct placement of CTb injection, or expression of jGCaMP or DREADD, within anterior insula were removed from the experiment.

## Apparatus

All procedures were performed in standard Med Associates operant chambers with data collected through the MED-PC IV program (Med Associates, Georgia, VT). Each chamber had one 'active' and one 'inactive' nose-poke hole on one wall and a grid floor connected to shock controllers. Contexts A and B were defined by houselight (on/off), cue-light color (white/red), and white noise (on/off).

The catheter for intravenous nicotine delivery was composed of a cannula connector pedestal (Plastics One, Minneapolis, MN), attached to a 95-mm silicone catheter (BC-2S; 0.3 mm × 0.6 mm; UNO B.V., Zevenaar, The Netherlands) and a 6-mm piece of polyethylene tubing (0.75 mm × 1.45 mm; UNO B.V., Zevenaar, The Netherlands) clamping the silicone catheter to the connector pedestal. A small ball of silicone (RTV-1 Silicone Rubber/Elastosil ) is attached 38 mm from the end of the silicone catheter.

For fiber photometry, excitation and emission light was relayed to and from the animal via optical fiber patch cord (0.48 NA, 400-μm flat tip; Doric Lenses). Blue excitation light (490 or 470 nm LED [M490F2 or M470F2, Thorlabs]) was modulated at 211 Hz and passed through a 460–490 nm filter (Doric Lenses), while isosbestic light (405 nm LED [M405F1, Thorlabs]) was modulated at 531 Hz and passed through a filter cube (Doric Lenses). GCaMP7f fluorescence was passed through a 500- to 550-nm emission filter (Doric Lenses) and onto a photoreceiver (Newport 2151). Light intensity at the tip of the fiber was measured before every training session and kept at 21 μW. A real-time processor (RZ5P, Tucker Davis Technologies) controlled excitation lights, demodulated fluorescence signals and received timestamps of behavioral events. Data were saved at 1017.25 Hz and analyzed with custom-made Matlab scripts, available at https://github.com/ialozares/NicInsulaPhotometry, copy archived at swh:1:rev:6d3efefa0f7e379c2f5bdccc1f3ca70d225bce5e (*Alonso-Lozares, 2022*).

## Drugs

Nicotine (nicotine hydrogen tartrate salt, Sigma-Aldrich, St. Louis, MO) was dissolved in normal saline, filtered, and pH adjusted to 7.4. CLZ was dissolved first in a small amount of glacial acetic acid (volume used was 0.1% of the final CLZ volume) and progressively diluted in saline until a final concentration of 0.3 mg/ml (pH was adjusted to 7.0–7.2).

## Viral vectors

We purchased premade viral vectors from the University of Zurich viral vector core: AAV-5/2-mCaMKIIa-HA_hM4D(Gi)-IRES-mCitrine-WPRE-hGHp(A) (**hM4Di**), AAV-5/2-mCaMKIIa-EGFP-WPRE-hGHp(A) (**GFP**), and AAV-9/2-hSyn1-chI-jGCaMP7f-WPRE-SV40p(A) (**jGCaMP7f**). The titer injected was: hM4Di, $2.4 \times 10^{12}$ gc/ml; GFP, $2.5 \times 10^{12}$ gc/ml; jGCaMP7f, $4.4 \times 10^{12}$ gc/ml.

## Surgery

Thirty minutes prior to surgery, we injected rats with the analgesic Rymadil (5 mg/kg; Merial, Velserbroek, The Netherlands) and the antibiotic Baytril (8.33 mg/kg; Bayer, Mijdrecht). Surgery was performed under isoflurane gas anesthesia (PCH; Haarlem). The silicone catheter was tunneled from the scalp to the neck and was inserted into the jugular vein, where it was secured using sterile thread. We sealed the silicone catheter using a taurolidine-citrate solution (Access Technologies, Skokie, IL) and a polyethylene cap. After the catheter was implanted, we placed the rat in a stereotactic frame (David Kopf Instruments, Tujunga, CA) and injected xylocaine 2% with adrenaline (10 mg/kg; Astra Zeneca, Zoetermeer, The Netherlands) into the incision site prior to the incision. A craniotomy above aIC was performed, followed by CTb or AAV injections (see below for details). After filling the skull hole with bone wax, cannula tubing connected to a Plastics One Connector-Pedestal and optic fiber implant (when applicable) was secured to the skull using dental cement (IV Tetric EvoFlow 2g A1,

Henry Schein, Almere) and jewelers screws. Rymadil (5 mg/kg; s.c.) was administered for 2 days after the surgery. Rats were given 1 week of recovery following surgery.

## CTb injections
40 nl of 1% CTb (List Biological Laboratories) was injected unilaterally (left or right) into aIC (AP: +2.8, ML: +4.0, DV: −5.9 mm from Bregma) over 2 min using 1.0 μl 32 gauge 'Neuros' syringe (Hamilton) attached to a UltraMicroPump (UMP3) with SYS-Micro4 Controller (World Precision Instruments). The needle was left in place for an additional 2 min after injections.

## AAV injections for fiber photometry
0.5 μl of AAV solution was injected unilaterally (left or right) into aIC (AP: +2.8, ML: +4.0, DV: −6.0 mm from Bregma) over 5 min. The needle was left in place for an additional 5 min. A 400-μm optic fiber (Doric Lenses) was then implanted above aIC (AP: +2.8, ML: +4.0, DV: −5.6 mm from Bregma).

## AAV injections for chemogenetics
1.0 μl of AAV solution was injected bilaterally into aIC (AP: +2.8, ML: +4.0, DV: −5.9 mm from Bregma) over 5 min using 10 μl Nanofil syringes (World Precision Instruments), with 33-gauge needles, attached to a UltraMicroPump (UMP3) with SYS-Micro4 Controller (World Precision Instruments). The needle was left in place for an additional 5 min.

## Behavioral procedure
### Phase 1: nicotine self-administration (SA: context A)
On the day prior to self-administration, before and after each self-administration session and during weekends, rat's catheters were flushed with approx. 0.1 ml mixture of heparin (0.25 mg/ml; Serva, Heidelberg, Germany) and gentamicin sulfate (0.08 mg/ml; Serva, Heidelberg, Germany). Rats were trained to self-administer nicotine in 2-hr sessions, 5 days a week. Entry into the active nose-poke resulted in intravenous nicotine delivery infused over approximately 2 s (infusion time adjusted for weight) at 30 μg/kg/infusion. Nicotine infusion was paired with a 20-s time-out period with the cue-light on. During time-out, responses were recorded but had no consequence. Inactive nose-pokes had no effect in either context. Rats were first trained on a fixed-ratio (FR) 1 schedule, which was then increased to FR2, followed by FR3. We tested catheter patency using intravenous anesthetic 0.05 cc pentothal (thiopenthal sodium, 50 mg/ml).

### Phase 2A: punishment of nicotine self-administration (PUN, context B)
Nicotine self-administration was maintained on the FR3 schedule, and 50% of the reinforced active nose-pokes (pseudo-randomly determined by the Med-PC program) resulted in footshock (0.30 mA for 0.5 s) and nicotine infusion.

### Phase 2B: extinction of nicotine-seeking (EXT, context B)
Entry into the active nose-poke resulted in simultaneous activation of the cue-light and delivery of the same volume of saline through the jugular catheter (FR3 schedule).

### Phase 3: relapse tests in context A (SA) and context B (PUN or EXT)
Following abstinence (PUN or EXT), rats were tested in contexts A and B. The response-contingent CS was presented on an FR3 schedule without punishment, saline or nicotine delivery. For the CTb +Fos experiment (Exp. 1), rats were tested for one 60-min session and perfused 90 min after the beginning of the test. For the chemogenetics experiments (Exp. 3 and 4) rats were tested in both contexts A and B (counterbalanced order). 30 min prior to the relapse test sessions, we injected both GFP and hM4Di expressing rats with CLZ at a dose of 0.3 mg/kg injection (i.p.). To habituate the rats to an i.p. injection, we injected approx. 0.1 ml saline i.p. prior to two nicotine self-administration sessions in context A, and two abstinence sessions (punish or extinction) in context B.

## Immunohistochemistry

We deeply anesthetized rats with isoflurane and Euthasol injection (i.p.) and transcardially perfused them with ~100 ml of normal saline followed by ~400 ml of 4% paraformaldehyde in 0.1 M sodium phosphate (pH 7.4). The brains were removed and postfixed for 2 hr, and then 30% sucrose in 0.1 M phosphate-buffered saline (PBS) for 48 hr at 4°C. Brains were then frozen on dry ice, and coronal sections were cut (40 μm) using a Leica Microsystems cryostat and stored in 0.1 M PBS containing 1% sodium azide at 4°C.

Immunohistochemical procedures are based on our previously published work (*Marchant et al., 2016*; *Marchant et al., 2014*; *Marchant et al., 2009*). We selected a 1-in-4 series and first rinsed free-floating sections (3× 10 min) before incubation in PBS containing 0.5% Triton-X and 10% Normal Donkey Serum (NDS) and incubated for at least 48 hr at 4°C in primary antibody. Sections were then repeatedly washed with PBS and incubated for 2–4 hr in PBS + 0.5% Triton-X with 2% NDS and secondary antibody. After another series of washes in PBS, slices were stained with 4',6-diamidino-2-phenylindole (DAPI; 0.1 μg/ml) for 10 min prior to mounting onto gelatin-coated glass slides, air-drying and cover-slipping with Mowiol and DABCO.

### CTb + Fos protein labeling

Primary antibodies were rabbit anti-c-Fos (1:2000; Cell Signaling, CST5348S) and goat anti-CTb (1:5000; List Biological Laboratories, 703). Secondary antibodies were donkey anti-rabbit Alexa Fluor 594 (1:500; Molecular Probes, A21207) and donkey anti-goat Alexa Fluor 488 (1:500: Molecular Probes, A11055).

### Photometry experiment

Primary antibodies were mouse anti-NeuN primary antibody (1:1000; Chemicon, MAB377) and rabbit anti-GFP primary antibody (1:2000; Chemicon, AB3080). Secondary antibodies were donkey anti-mouse DyLight 649 (1:500; Jackson ImmunoResearch, 715-495-150) and donkey anti-rabbit Alexa Fluor 594 (1:500: Mol. Probes, A21207).

### Chemogenetic inhibition experiments

Primary antibody was rabbit anti-GFP primary antibody (1:2000; Chemicon, AB 3080) and secondary antibody was donkey anti-rabbit Alexa Fluor 594 (1:2000; Molecular Probes, A21207). In rats in the GFP group, slices were only stained with DAPI.

## Image acquisition and neuronal quantification

Slides were all imaged on a VectraPolaris slide scanner (VUmc imaging core) at ×10 magnification. For the CTb + Fos experiment (Exp. 1), images from Bregma +4.0 to Bregma −3.3 were scanned and imported into QuPath for analysis (*Bankhead et al., 2017*). For photometry and chemogenetic experiments, images containing aIC, from Bregma +4.2 mm to +2.5 mm were identified and the boundary of expression for each rat was plotted onto the respective (*Paxinos and Watson, 2008*). Rats in Exp. 1 that had a CTb injection not within aIC were excluded from analysis. Rats in Exp. 4 and 5 that had either unilateral expression or misplaced expression were excluded from the analysis.

### Fos, CTb, and CTb + Fos quantification

Regions of interest were manually labeled across sections using DAPI for identification of anatomical landmarks and boundaries. For the IC, we labeled 18 sections per hemisphere per rat spaced approximately 400 μm apart. For analysis, we separated IC into three regions anterior (aIC), middle (mIC), and posterior (pIC), and each value was the result of an average of each count from six adjacent sections: aIC (approx. Bregma + 3.72 to+ 1.44), mIC (approx. Bregma +1.08 to −0.72), pIC (approx. Bregma −1.08 to −2.92). For BLA, we labeled six sections spaced approximately 200 μm apart, and we separated it into anterior BLA (aBLA) and posterior BLA (pBLA), which is the average value of three adjacent sections: aBLA (approx. Bregma −1.92 to −2.40), pBLA (approx. Bregma −2.64 to −3.12). Some rats had missing sections due to mistakes during the process, and these sections were left blank for the statistical analyses. The contralateral BLA was not counted because we found no CTb-labeled projections from contralateral BLA into aIC.

To identify Fos- and CTb-positive cells, we used the 'Cell detection' feature in QuPath, with an identical threshold applied across all sections. CTb was not counted for the first six sections in the ipsilateral aIC, where the CTb injection was located, because the cell detection feature could not reliably discriminate between CTb-positive cell and the CTb injection. The total number of positive cells per region was divided by the area in mm². To identify CTb + Fos cells, each region of interest was exported to ImageJ. The overlays representing the cells (CTb or Fos) were then filled, converted to a binary layer, and then multiplied using the ImageJ function 'Image calculator'. The nuclei that remained as a result of this function were counted as double-labeled CTb + Fos neurons. CTb + Fos double labelling is reported as a percentage of total CTb neurons for that given region of interest.

## Ex vivo slice physiology

Coronal slices were prepared for electrophysiological recordings. Rats were anesthetized (5% isoflurane, i.p. injection of 0.1 ml/g pentobarbital) and perfused with ice-cold $N$-methyl-D-glucamin (NMDG) solution containing (in mM): NMDG 93, KCl 2.5, $NaH_2PO_4$ 1.2, $NaHCO_3$ 30, 4-(2-hydroxyethyl)-1-piperazineethanesulfonic acid (HEPES) 20, glucose 25, sodium ascorbate 5, sodium pyruvate 3, $MgSO_4·2H_2O$ 10, $CaCl_2 · 2H_2O$ 0.5, at pH 7.3 adjusted with 10 M HCl. The brains were removed and incubated in ice-cold NMDG solution. 300 μm thick brain slices were cut in ice-cold NMDG solution and subsequently incubated for 15–30 min at 34°C.

Before the start of experiments, slices were allowed to recover for at least 1 hr at room temperature in carbogenated (95% $O_2$/5% $CO_2$) ACSF solution containing (in mM): NaCl 125, KCl 3, $NaH_2PO_4$ 1.2, $NaHCO_3$ 25, glucose 10, $CaCl_2$ 2, $MgSO_4$ 1. For voltage- and current-clamp experiments borosilicate glass patch-pipettes (3–5 MΩ) were used with a K-gluconate-based internal solution containing (in mM): K-gluconate 135, NaCl 4, MgATP 2, phosphocreatine 10, GTP (sodium salt) 0.3, EGTA 0.2, HEPES 10 at a pH of 7.4. Data were sampled using a Multiclamp 700B amplifier (Axon Instruments) and pClamp software (Molecular Devices). All recordings were made between 31.1 and 33.6°C.

## Experimental design

### Exp. 1: CTb + Fos after context-induced relapse of punished nicotine-seeking ($n$ = 18 female; cohort 1, $n$ = 11; cohort 2, $n$ = 7)

*Figure 1A* shows the experimental outline. We first trained rats to self-administer nicotine in one context (context A) in 2 hr sessions per day for 15 days (four sessions FR1, six sessions FR2, five sessions FR3). Next, the rats underwent punishment in the alternate context (context B) 2 hr per day for 7 days. During these sessions, active nose-pokes (FR3 schedule) resulted in the presentation of the cue-light (20 s), nicotine infusion, and 50% probability of 0.3 mA footshock punishment. Finally, the rats were tested under extinction conditions. One group of rats ($n$ = 7) was tested in the nicotine self-administration context (context A), one group ($n$ = 8) was tested in the punishment context (context B), and a third group ($n$ = 3) was taken from the home-cage without test. We perfused rats 90 min after the start of the 60-min test. We excluded 1 rat from the experiment due to a lack of CTb expression.

### Exp. 2: calcium imaging of aIC activity during nicotine self-administration, punishment, and context-induced relapse ($n$ = 7 female; cohort 1, $n$ = 3; cohort 2, $n$ = 4)

*Figure 2A* shows the experimental outline. Photometry sessions were 1 hr in duration, and we recorded every session throughout this experiment. Tubing delivering nicotine was run down along the fiber-optic patch-cord and to the implanted optic fiber. If the tubing became tangled, the rat was manually rotated in the opposite direction or, if it was within 10 min of the end of the session, the session was ended early. We gave no prior habituation sessions, and we found that nicotine self-administration was at a comparable rate in recording session relative to experiments without photometry recording. Rats were trained to self-administer nicotine in context A (7 days FR1, 3 days FR2, 8 days FR3). We next punished nicotine self-administration in context B for 6 days (first cohort; $n$ = 3) or 10 days (second cohort; $n$ = 3), one rat only received 4 days of punishment. In this phase 50% of nicotine infusions were paired with electric shock. After punishment-imposed abstinence, nicotine-seeking was tested in context A in 2 × 1-hr sessions over 2 consecutive days. We did not perform a

context B test because we did not expect there to be enough events (i.e., nose-pokes) to obtain a reliable aIC activity readout.

## Exp. 3: chemogenetic validation (n = 3 females, n = 3 males)

Six rats were unilaterally injected with 1.0 µl of AAV encoding the inhibitory DREADD hM4Di into aIC (AP: +2.8, ML: +4.0, DV: −5.9 mm from Bregma). Rats were sacrificed 4–5 weeks later for ex vivo physiology.

## Exp. 4: effect of chemogenetic inhibition of aIC on context-induced relapse of punished nicotine-seeking (n = 28 [13M/15F]; cohort 1, n = 17 [8M/9F]; cohort 2, n = 11 [5M/6F])

*Figure 4A* shows the experimental outline. We first trained rats to self-administer nicotine in context A (4 days FR1, 4 days FR2, 5 days FR-3). We next punished nicotine self-administration in context B for 6 days. We then tested rats in both contexts (A and B), over 2 consecutive days, and the order was counterbalanced. 30 min prior to the test session, we injected both GFP and hM4Di expressing rats with CLZ 0.3 mg/kg injection (i.p.). We excluded three female rats from the hM4Di group because of a lack of bilateral hM4Di expression in aIC.

## Exp. 5: effect of chemogenetic inhibition of aIC on context-induced relapse of extinguished nicotine-seeking (n = 31 [13M/18F]; cohort 1, n = 22 [11M/11F]; cohort 2, n = 9 [2M/7F])

*Figure 5A* shows the experimental outline. We first trained rats to self-administer nicotine in context A on FR1 (4 days), then FR2 (4 days), then FR-3 (5 days). Next, we extinguished nicotine-seeking by saline infusion in context B (EXT) for 8 days. We then tested rats in both contexts (A and B), over 2 consecutive days, and the order was counterbalanced. 30 min prior to the test session, we injected both GFP and hM4Di expressing rats with CLZ 0.3 mg/kg injection (i.p.).

## Statistics

All behavioral data were analyzed using IBM SPSS V21. Phases were analyzed separately. Dependent variables were the total number of active and inactive nose-pokes across phases, and nicotine infusions for nicotine self-administration and punishment phases. For the CTb + Fos test (Exp. 1) we used a repeated measures ANOVA with Nose-Poke (Active, Inactive) as a within-subjects factor and Test Context (contexts A and B) as the between-subjects factor. To analyze CTb + Fos expression, we used one-way ANOVA to test for an effect of Test Context (Home-cage, Punishment, Nicotine) on Fos, CTb, and % CTb + Fos/CTb. Follow-up tests (Tukey) were conducted on regions that had a significant main effect of Test Context. For the chemogenetic experiments we used repeated measures ANOVA with Test Context (contexts A and B) and Nose-Poke (Active, Inactive) as within-subjects factors, and Virus (GFP, hM4Di) and Sex (Female, Male) as between-subjects factors.

### Photometry

Recorded signals were first downsampled by a factor of 64, giving a final sampling rate of 15.89 Hz. The 405 nm isosbestic signal was fit to the 490 nm calcium-dependent signal using a first-order polynomial regression. A normalized, motion-artifact-corrected $\Delta F/F$ was then calculated as follows: $\Delta F/F$ = (490 nm signal − fitted 405 nm signal)/fitted 405 nm signal. The resulting $\Delta F/F$ was then detrended via a 90-s moving average, and low-pass filtered at 3 Hz. $\Delta F/F$ from 5 s before nose-poke (baseline) to 10 s after nose-poke were c-ollated. These traces were then baseline corrected and converted into z-scores by subtracting the mean baseline activity during first 4 s of the baseline and dividing by the standard deviation of those 4 s. To avoid duplicate traces due to overlapping epochs, we excluded from the analyses any nose-pokes that occurred within 20 s after a rewarded nose-poke (i.e., all time-out responses), and unrewarded active/inactive nose-pokes that occurred 5 s after another active/inactive nose-poke.

Nose-poke traces were grouped by response type: active rewarded, active punished, active nonrewarded, and inactive. Two analysis approaches were used, bootstrapping and permutation tests, the

rationale for each is described in detail in *Jean-Richard-Dit-Bressel et al., 2020*. The output of every statistical test we conducted is available in the raw data files.

Bootstrapping was used to determine whether calcium activity per response type was significantly different from baseline ($\Delta F/F = 0$). A distribution of bootstrapped means were obtained by randomly sampling from traces with replacement (*n* traces for that response type; 5000 iterations). A 95% confidence interval was obtained from the 2.5th and 97.5th percentiles of the bootstrap distribution, which was then expanded by a factor of sqrt($n/(n − 1)$) to account for narrowness bias (*Jean-Richard-Dit-Bressel et al., 2020*).

Permutation tests were used to assess significant differences in calcium activity between response types. Observed differences between response types were compared against a distribution of 1000 random permutations (difference between randomly regrouped traces) to obtain a p value per time point. Alpha of 0.05 was Bonferroni corrected based on the number of comparison conditions, resulting in alpha of 0.01 for comparisons between self-administration and test sessions (three conditions) and alpha of 0.008 for punishment sessions (four conditions). For both bootstrap and permutation tests, only periods that were continuously significant for at least 0.25 s were identified as significant (*Jean-Richard-Dit-Bressel et al., 2020*).

For the mean d$f$/$f$ presentation of the data (*Figure 2—figure supplement 2*) we calculated the mean d$f$/$f$ for the 2 s prior to the nose-poke ('−2 → 0 s') and the 5 s after the response ('0 → 5 s'). These values were then compared in a within-subjects *t*-test to determine whether there was a significant change in activity in this specific time window.

## Acknowledgements

The authors gratefully acknowledge the VUmc Histology Imaging Unit for their support & assistance in whole-slide imaging. The authors would like to thank Francesco Ferraguti for insightful comments in the preparation of this manuscript. The work was supported by an NWO VIDI grant (016.Vidi.188.022), Fulbright Fellowship to RH, and Austrian Science Fund (FWF) grant Signal Processing in Neurons (SPIN) W1206-12 to HG and GZ (graduate program Signal Processing In Neurons, https://www.neurospin.at/). The authors declare no conflict of interest.

## Additional information

### Funding

| Funder | Grant reference number | Author |
| --- | --- | --- |
| Nederlandse Organisatie voor Wetenschappelijk Onderzoek | 016.Vidi.188.022 | Nathan J Marchant |
| Fulbright Association | | Rae J Herman |
| Austrian Science Fund | Signal Processing in Neurons (SPIN) grant W1206-12 | Hussein Ghareh Gerald Zernig |

The funders had no role in study design, data collection, and interpretation, or the decision to submit the work for publication.

### Author contributions

Hussein Ghareh, Conceptualization, Formal analysis, Investigation, Methodology, Writing – original draft, Writing – review and editing; Isis Alonso-Lozares, Conceptualization, Data curation, Formal analysis, Methodology, Validation, Visualization, Writing – original draft, Writing – review and editing; Dustin Schetters, Tim S Heistek, Yvar Van Mourik, Investigation, Methodology, Writing – review and editing; Rae J Herman, Conceptualization, Data curation, Investigation, Methodology, Writing – review and editing; Philip Jean-Richard-dit-Bressel, Formal analysis, Methodology, Software, Visualization, Writing – review and editing; Gerald Zernig, Project administration, Supervision, Writing – review and editing; Huibert D Mansvelder, Funding acquisition, Project administration, Supervision, Writing – review and editing; Taco J De Vries, Conceptualization, Funding acquisition, Project administration,

Supervision, Writing – review and editing; Nathan J Marchant, Conceptualization, Data curation, Formal analysis, Funding acquisition, Methodology, Project administration, Supervision, Visualization, Writing – original draft, Writing – review and editing

## Author ORCIDs
Rae J Herman http://orcid.org/0000-0001-6119-3027
Gerald Zernig http://orcid.org/0000-0002-1247-1024
Huibert D Mansvelder http://orcid.org/0000-0003-1365-5340
Taco J De Vries http://orcid.org/0000-0002-0340-4946
Nathan J Marchant http://orcid.org/0000-0001-8269-0532

## Ethics
In compliance with Dutch law and Institutional regulations, all animal procedures were approved by the Centrale Commissie Dierproeven (CCD) and conducted in accordance with the Experiments on Animal Act. Experiments were approved by the local animal welfare body Animal Experiments Committee of the Vrije Universiteit, Amsterdam, The Netherlands.

## Decision letter and Author response
Decision letter https://doi.org/10.7554/eLife.75609.sa1
Author response https://doi.org/10.7554/eLife.75609.sa2

---

# Additional files

## Supplementary files
• Transparent reporting form

## Data availability
All data generated or analyzed during this study are included in the manuscript and supporting file; Source Data files have been provided for Figures 1-5, and Figure supplements 1-3.

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
