## [Editor Report]

This manuscript is of interest to readers in the fields of drug addiction and relapse, reinforcement learning and punishment, and those interested in cortical functions, particularly the insular cortex. The authors extend a context and punishment-based relapse model to the widely-used drug nicotine and use a number of carefully controlled complementary approaches ranging from chemogenetic interventions to fiber photometry to support the conclusion that the insular cortex plays a role in nicotine relapse.

---

## [Decision Letter]

**Decision letter after peer review:**

Thank you for submitting your article "Role of anterior insula cortex in context-induced relapse of nicotine-seeking" for consideration by *eLife*. Your article has been reviewed by 3 peer reviewers, and the evaluation has been overseen by a Reviewing Editor and Michael Taffe as the Senior Editor. The reviewers have opted to remain anonymous.

Essential revisions:

1) Further analyses and precisions are needed for the photometry approach. Individual reviews provided by all referees suggest multiple lines of improvements. While we recognize that it may be difficult to address each issue, it is expected that the paper is substantially strengthened here by providing statistical analyses, possibly new angles to look at the data and by adding on the discussion of the approach and how it complements with the DREADD experiments.

2) Given that no vehicle-treated groups were included in the study to formally rule out any off-site effect of clozapine, further caution may be needed to interpret the DREADD experiments. See referee 3 in particular.

3) The rationale for focusing on the contralateral aIC and ipsilateral BLA inputs needs to be more clearly explained.

*Reviewer #2 (Recommendations for the authors):*

1. In experiment 1 the authors focused their analysis specifically to contralateral aIC and ipsilateral BLA inputs. Why did the authors focused their analysis on these two specific inputs? Did the authors look at over inputs? In the same experiment, the quantification of CTb and fos staining has been performed in the anterior and posterior BLA. Are there any anatomical or functional justifications for this anteroposterior segregation?

2. The authors made a significant effort to characterize the coding properties of aIC neurons during nicotine self-administration, punishment, and context-induced relapse. However, several concerns are raised in the analysis of the calcium signal.

For each of the 3 phases of the task (Self-administration, punishment, relapse) the authors report a peri-event analysis of the calcium signal between pre and post nose-poke averaging the signal across the sessions. However, it cannot be ruled out that the calcium signal change between sessions (i.e: first and last self-administration session, first and second relapse test…) and within sessions (i.e: comparison between first and last nose-poke of self-administration session or relapse test). This deep analysis would strengthen the results and would bring a deeper understanding of the coding properties of aIC neurons between the different phases of the task. Finally, a correlational analysis between the calcium signal and the behavioral measures (i.e: Nose-pokes, nicotine intake…) would reinforce the conclusions.

3. In line with the previous comment, more details about the fibre photometry recording experiments would be helpful. Were the rats habituated to be tethered before to starting the procedure? State in the methods that the recording was performed every day of the task. Moreover, in the fibre photometry experiment the sessions of self-administration and punishment are reduced to 1h compared to the other experiments with 2-h/session. This difference suggests that the rats received less nicotine intake and context/cues exposure, it should be acknowledged. Did the authors compare the average nicotine intake between the experiments?

4. Concerning the representation of the calcium signal differences. To help the reader appreciate the variation of signal I would suggest representing the most relevant comparisons as a histogram of the average of the calcium transient (i.e calcium transient average before nose-poke for active NP Nic vs calcium transient after nose-poke for active NP Nic).

5. Although discussed in the paper one important limitation in the methodology and the interpretation of the results is the lack of consistency between the promoteor of the viral vector used. In the fibre photometry experiments, the authors used hSyn promoteor suggesting that they recorded the calcium signal from all neurons of the aIC (glutamatergic neurons and GABAergic interneurons) whereas in the chemogenetic experiments they used CaMKII promoteor allowing to specifically target the glutamatergic neurons. Fiber photometry recording using CaMKII promoteor would strengthen the results and the interpretation of the coding and causal properties of the aIC neurons.

6. The authors made an effort to include both male and female rats in their study. However, both sexes are not included in all the experiments making difficult to conclude about presence or lack of sex differences. It should be at least acknowledged.

*Reviewer #3 (Recommendations for the authors):*

This is an excellent paper and the authors should consider addressing the two points listed in the weaknesses section.

1. The authors should consider discussing the limitations of interpretation of the fibre photometry data, particularly in regards to alternative interpretations of the increase in activity in the active nose poke -> punished condition and how this may be due to the multiple outcomes (nicotine infusion + punishment). It maybe useful to indicate on the graphs when the footshock and nicotine infusions took place.

2. The authors should consider testing if there is an effect on responding with chemogenetic inhibition of the anterior insular cortex in a procedure where responding is high (e.g. nicotine or food self-administration). Interpretation of the current data in Context B could be confounded by a floor effect.

---

## [Author Response]

Essential revisions:1) Further analyses and precisions are needed for the photometry approach. Individual reviews provided by all referees suggest multiple lines of improvements. While we recognize that it may be difficult to address each issue, it is expected that the paper is substantially strengthened here by providing statistical analyses, possibly new angles to look at the data and by adding on the discussion of the approach and how it complements with the DREADD experiments.

In the revised manuscript we have provided more detailed descriptions of the statistical effects in the Results section. We have also included the raw output from the bootstrap analysis (confidence intervals) and permutation tests (p values) in the updated raw data files. We have also added some points in the discussion to better describe the link between the photometry data and the DREADD experiments, and also point out the strong overlap between the Fos data and these experiments.

Finally, we also performed two additional analyses on the photometry data (in the supplementary data). First, we show the photometry data from the different FR stages of self-admin (FR1, FR2, and FR3), the first punishment session, and the first and second relapse test. Second we calculated the mean df/f for the different events and plotted them to illustrate the individual variability between the rats in each phase.

2) Given that no vehicle-treated groups were included in the study to formally rule out any off-site effect of clozapine, further caution may be needed to interpret the DREADD experiments. See referee 3 in particular.

We agree that there is potential for non-specific effects of clozapine in our experiments, and have updated the discussion to address this. To further attempt to address this, we perform additional statistical analysis on the inactive nose-pokes and found no effect of clozapine, despite the rate of inactive nose-pokes being relatively high (going some way to rule out floor effects).

3) The rationale for focusing on the contralateral aIC and ipsilateral BLA inputs needs to be more clearly explained.

We have explained this rationale in the revised manuscript. We choose to focus on the most prominent inputs to the aIC, the more caudal regions of IC and the BLA. Only ipsilateral BLA was counted because we found no projections to aIC from contralateral BLA, which is consistent with the literature. We have revised the manuscript to more carefully explain these points.

Reviewer #2 (Recommendations for the authors):1. In experiment 1 the authors focused their analysis specifically to contralateral aIC and ipsilateral BLA inputs. Why did the authors focused their analysis on these two specific inputs? Did the authors look at over inputs? In the same experiment, the quantification of CTb and fos staining has been performed in the anterior and posterior BLA. Are there any anatomical or functional justifications for this anteroposterior segregation?

We have not focused on other inputs in this study. We report the anterior and posterior BLA because we counted 6 sections per rat in the AP axis (range approx. Bregma -1.92 to -3.12), and there are studies demonstrating differences in both anatomy and function in anterior versus posterior BLA. In the revised manuscript we describe some of these findings.

2. The authors made a significant effort to characterize the coding properties of aIC neurons during nicotine self-administration, punishment, and context-induced relapse. However, several concerns are raised in the analysis of the calcium signal.For each of the 3 phases of the task (Self-administration, punishment, relapse) the authors report a peri-event analysis of the calcium signal between pre and post nose-poke averaging the signal across the sessions. However, it cannot be ruled out that the calcium signal change between sessions (i.e: first and last self-administration session, first and second relapse test…) and within sessions (i.e: comparison between first and last nose-poke of self-administration session or relapse test). This deep analysis would strengthen the results and would bring a deeper understanding of the coding properties of aIC neurons between the different phases of the task. Finally, a correlational analysis between the calcium signal and the behavioral measures (i.e: Nose-pokes, nicotine intake…) would reinforce the conclusions.

We agree with the reviewer that there are many potential advantages to a more refined analysis. In the revised manuscript we report analysis of the different FR stages of self-admin (FR1, FR2, and FR3), as well as the first punishment session, and the first and second relapse test. Due to the variability of the data, and statistical limitations, we cannot go further than this, for example looking within sessions (e.g. first vs. last nose-poke), or the last punishment session because of the low number of recorded events.

We think these new analyses identify a number of interesting patterns. First in self-administration we show that the response to nicotine is present in the first sessions (FR1), and remains present throughout self-administration. Second, we show that the response to punishment dramatically overshadows that of nicotine initially, and this response to punishment declines as behaviour decreases (with the limitation of a smaller number of events in the last punishment session). Also in punishment we show that the response to nicotine infusions is no longer significant above baseline in the first session. Third we show that the selective increase in activity prior to a response (i.e. active versus inactive) is only present in the first test session.

3. In line with the previous comment, more details about the fibre photometry recording experiments would be helpful. Were the rats habituated to be tethered before to starting the procedure? State in the methods that the recording was performed every day of the task. Moreover, in the fibre photometry experiment the sessions of self-administration and punishment are reduced to 1h compared to the other experiments with 2-h/session. This difference suggests that the rats received less nicotine intake and context/cues exposure, it should be acknowledged. Did the authors compare the average nicotine intake between the experiments?

We appreciate this comment, and have added information about this in the Methods in the revised manuscript.

4. Concerning the representation of the calcium signal differences. To help the reader appreciate the variation of signal I would suggest representing the most relevant comparisons as a histogram of the average of the calcium transient (i.e calcium transient average before nose-poke for active NP Nic vs calcium transient after nose-poke for active NP Nic).

We have added Figure 2—figure supplement 2 showing the calcium transient calculated using mean df/f for the 2 second preceding the nose-poke and 5 seconds after. We do this for every data trace that we reported in the main figure (2).

5. Although discussed in the paper one important limitation in the methodology and the interpretation of the results is the lack of consistency between the promoteor of the viral vector used. In the fibre photometry experiments, the authors used hSyn promoteor suggesting that they recorded the calcium signal from all neurons of the aIC (glutamatergic neurons and GABAergic interneurons) whereas in the chemogenetic experiments they used CaMKII promoteor allowing to specifically target the glutamatergic neurons. Fiber photometry recording using CaMKII promoteor would strengthen the results and the interpretation of the coding and causal properties of the aIC neurons.

We appreciate this comment. In the revised discussion we raise this issue and discuss potential future directions related to these point

6. The authors made an effort to include both male and female rats in their study. However, both sexes are not included in all the experiments making difficult to conclude about presence or lack of sex differences. It should be at least acknowledged.

We have acknowledged this point in the revised manuscript.

Reviewer #3 (Recommendations for the authors):This is an excellent paper and the authors should consider addressing the two points listed in the weaknesses section..1. The authors should consider discussing the limitations of interpretation of the fibre photometry data, particularly in regards to alternative interpretations of the increase in activity in the active nose poke -> punished condition and how this may be due to the multiple outcomes (nicotine infusion + punishment). It maybe useful to indicate on the graphs when the footshock and nicotine infusions took place.

We thank the reviewer for this comment and have changed the figures to more clearly show the experimental design.

2. The authors should consider testing if there is an effect on responding with chemogenetic inhibition of the anterior insular cortex in a procedure where responding is high (e.g. nicotine or food self-administration). Interpretation of the current data in Context B could be confounded by a floor effect.

To address this potential issue, we conducted additional analysis on the latency to the first active and inactive nosepoke (Figure 4—figure supplement 1; Figure 5—figure supplement 1) which shows that there was no difference between inhibited aIC and control. In the discussion we argue that this observation further illustrates the specificity of the effect of aIC inhibition of nicotine seeking.